# Analysis of the Small Auxin-Up RNA (SAUR) Genes Regulating Root Growth Angle (RGA) in Apple

**DOI:** 10.3390/genes13112121

**Published:** 2022-11-15

**Authors:** Yufei Zhou, Qingqing Lan, Wenhan Yu, Yuwen Zhou, Shuya Ma, Zeyang Bao, Xu Li, Caixia Zheng

**Affiliations:** 1College of Biological and Environmental Sciences, Zhejiang Wanli University, Ningbo 315100, China; 2College of Agriculture and Bioengineering, Heze University, Heze 274000, China; 3College of Food Science and Engineering, Ocean University of China, Qingdao 266003, China

**Keywords:** apple, root growth angle, SAURs, expression profiles, allelic variants

## Abstract

Small auxin upregulated RNAs (SAURs) are primary auxin response genes; the function of regulating root growth angle (RGA) is unclear in the apple rootstock. We firstly identified 96 MdSAUR genes families from new apple genome GDDH13 using the resequence database of ‘Baleng Crab (BC)’ and ‘M9’. A total of 25 MdSAUR genes, regulating the formation of RGA, were screened for the expression profiles in stems and roots and the allelic variants of quantitative trait loci (QTL). Finally, through the joint analysis of network and protein–protein interaction, *MdSAUR2*, *MdSAUR29*, *MdSAUR60*, *MdSAUR62*, *MdSAUR69*, *MdSAUR71*, and *MdSAUR84* were screened as the main candidate genes for regulating RGA. This study provides a new insight for further revealing the regulatory mechanism of RGA in apple dwarf rootstocks.

## 1. Introduction

Apple (*Malus domestica* Borkh.) is one of the most popular perennial tree fruits in the world [1]. Currently, the dwarf form and close planting is the main cultivation mode, which is used to develop the apple industry. However, dwarfing rootstock is the limiting factor that restricts the popularization of dwarf and close planting [2]. An ideal dwarfing rootstock with a relatively deep root system is pivotal for water uptake, nutrient uptake, and adaptability [3]. Root growth angle (RGA), defined as the angle between the direction of root growth and the horizontal axis, determines the depth of root architecture and the development of high-density planting of apple [4]. The bending of RGA results from differential elongation of the cells in the distal elongation zone, which results from the differential accumulation of auxins on opposite sides of the root [5,6]. However, the underlying mechanisms of auxin in RGA is unclear.

Auxin mainly includes three different types of early auxin-responsive gene families, namely, Aux/indole-3-acetic acid (IAA), Gretchen Hagen 3 (GH3), and small auxin-up RNA (SAUR) [7]. Auxin signal transduction by numerous transcription factors that act downstream of IAA and auxin response factor (ARF) proteins leads to a variety of auxin-induced morphological changes in plants, including gravitropic bending [8,9]. Precise auxin levels are vitally important to plants, which have many effective mechanisms to maintain auxin homeostasis [10]. The GH3 gene family, a supervisor of the fluctuation of auxin, could maintain auxin homeostasis by a mechanism in which an amino acid of some GH3 proteins conjugates to excessive IAA and inactivates IAA [11]. The SAUR gene family comprises the most rapid auxin-responsive genes related to the auxin signaling pathway [7,12]. The expression of SAURs can be induced within minutes by the active auxin, implying that auxin plays a critical role in their gene transcription [13]. However, many SAURs are regulated post-transcriptionally as a conserved downstream destabilizing element in the 3′-untranslated region that confers high mRNA instability [14]. Therefore, regulation of SAURs may occur at the transcriptional, posttranscriptional, and protein levels [15,16].

After the first SAUR gene was identified in soybean [17], members of SAUR gene family have been identified in a wide range of plants such as rice [18], moso bamboo [19], and poplar [20]. In apple, a total of 80 MdSAUR genes have been successfully identified in the draft of the apple genome [21,22]. Now, a new, high-quality, de novo assembly of the apple genome, named GDDH13, has been updated [23]. More accurate gene sequences and different designations were identified in the new apple genome GDDH13 [24]. Therefore, it is necessary to further identify the *MdSAUR* gene family in this new apple genome. In our previous study, the BSA-seq and RNA-seq of hybrids from ‘M9’ and ‘BC’ were analyzed, which contributes to further screening of MdSAUR genes regulating RGA from expression and allelic variation [25].

In this study, a genome-wide analysis for the homologues SAUR genes in the new apple genome, GDDH13, was carried out. Subsequently, the expression profiles of MdSAURs in the stems and roots of ‘M9’ and ‘BC’ after cutting were measured by quantitative real-time PCR (qRT-PCR). Then, the allelic variants of MdSAURs located in QTLs were predicted by the resequence database of ‘M9’ and ‘BC’. Finally, these network analyses of MdSAURs were analyzed to further screen MdSAURs genes regulating RGA.

## 2. Results

### 2.1. Identification of SAUR Gene Family in Apple Genome GDDH13

According the resequence database of ‘BC’ and ‘M9’, a total of 96 *SAUR* genes in the new apple genome GDDH13 were identified and designated as *MdSAUR* 1–96, ordered by their location in the chromosome from the top to bottom of the apple chromosomes 0–17 (Table 1). The number of *MdSAUR* genes per chromosome ranged from 1 (chromosome 6, 11, 17) to 25 (chromosome 10). The protein lengths of coding sequences range from 51 (*MdSAUR 5*) to 219 amino acids (*MdSAUR 71*), with an average sequence length of 124 amino acids. The lengths of *MdSAUR* genes in the genome vary from 156 to 660 bp. According to the results of the predicted protein localization, we found that 93 SAUR proteins possess signal sequences targeting in the cytoplasmic. *MdSAUR 33* and *MdSAUR 71* were only located in periplasmic. *MdSAUR 56* was in the outer membrane and periplasmic.

### 2.2. Phylogenetic Relationships of SAUR Genes in Arabidopsis and Apple

A phylogenetic tree was constructed on the basis of 231 putative nonredundant SAUR protein sequences from *Arabidopsis*, rice, and apple. All SAUR proteins were clustered into five groups (Figure 1, Appendix A). The number of SAUR proteins per group ranged from 26 (group V) to 64 (group I). Groups II, III, and V contained more SAUR proteins in apple than in *Arabidopsis* and rice, whereas the opposite was found in group I. Group IV contained 14 MdSAUR proteins, which was more than the number of OsSAURs (9), and less than the number of AtSAURs (20).

### 2.3. Expression Analysis of MdSAUR Genes in Stems and Roots

A total of 36 *MdSAURs* genes had no expression level in stems of hybrids from ‘M9’ and ‘BC’ at 7 d after cutting by RNA-seq analysis (Appendix A). Of 60 expressed *MdSAURs*, 18 had a higher transcripton level in hybrids of large RGA than those of small RGA (Appendix A; Figure 2a). At 14 d after cutting, roots of hybrids from ‘M9’ and ‘BC’ owned 53 expressed *MdSAURs*, but didn’t contain the different expression genes (Appendix A; Figure 2b). These candidate *MdSAUR* genes were further screened by using qRT-PCR analysis of ‘M9’ and ‘BC’ (Figure 3). The expression level of 18 *MdSAUR* genes in stems of ‘BC’ (large RGA) was higher than in those of ‘M9’ (small RGA). In roots of ‘M9’ and ‘BC’, the expression of the 18 *MdSAURs* had no different expression levels.

### 2.4. Co-Localization of MdSAURs with QTL for RGA

To better understand the potential function of SAUR genes related to RGA, we co-localized the SAUR genes with reported RGA QTL. As a result, a total of 24 *MdSAUR* genes were mapped in QTL M10.1 and B13.2 by BAS-seq analysis of hybrids with large–small RGA from ‘BC’ and ‘M9’ (Figure 4). To understand if the co-localized SAURs are genetically associated with RGA, allelic variations were analyzed to gain an insight into the *MdSAUR* genes in QTLs. In the QTL M10.1, of 23 *MdSAURs*, there were 12 *MdSAUR* genes existing allelic variations. *MdSAUR84* was located in QTL B13.2 and had allelic variations. Of 13 MdSAURs, the allelic variations of *MdSAUR56*, *MdSAUR65*, *MdSAUR67*, *MdSAUR69*, and *MdSAUR71* were only in the CDS. In addition, *MdSAUR55*, *MdSAUR61*, and *MdSAUR68* were variations in the promoter. Variations of *MdSAUR53*, *MdSAUR57*, *MdSAUR58*, *MdSAUR62*, and *MdSAUR84* were in both the CDS and promoter. Among those *MdSAUR* genes, *MdSAUR57* contained insertion variations, *MdSAUR71* had deletion variations, and other 11 *MdSAURs* genes only had SNP allelic variations. The exon–intron structure showed that only *MdSAUR69* had an intron (Figure 5). Exons of *MdSAUR58* and *MdSAUR69* contained CDS and UTR, and other 11 *MdSAUR* genes were only the CDS (Figure 5).

### 2.5. Joint Screening of Candidate MdSAURs Regulating RGA

By analysis of expression and allelic variations, 25 MdSAURs regulating RGA formation were screened. Eight genes (*MdSAUR2*, *MdSAUR14*, *MdSAUR17*, *MdSAUR18*, *MdSAUR19*, *MdSAUR21*, *MdSAUR27*, and *MdSAUR29*) were only differential expression genes, seven genes (*MdSAUR55*, *MdSAUR65*, *MdSAUR68*, *MdSAUR69*, *MdSAUR71*, *MdSAUR72*, and *MdSAUR84*) were only mapped in QTL, and another 10 genes (*MdSAUR53*, *MdSAUR56*, *MdSAUR57*, *MdSAUR58*, *MdSAUR59*, *MdSAUR60*, *MdSAUR61*, *MdSAUR62*, *MdSAUR67*, and *MdSAUR70*) were both differential expression genes and mapped in QTL (Figure 6).

### 2.6. Network Analysis of Candidate MdSAURs

We predicted potential proteins which may regulate 25 candidate MdSAUR genes by network analysis in the AppleMDO online database. Based on the Pearson correlation coefficient (PCC) ≥ 0.9 and the numbers of networks ≥ 10, six MdSAURs were predicted as primary candidate genes (Figure 7, Appendix A). The numbers of potential proteins with MdSAURs ranged from 21 (*MdSAUR2*) to 67 (*MdSAUR71*). *MdSAUR2*, *MdSAUR69*, and *MdSAUR84* formed three independent networks with different proteins of 21, 31, and 10, respectively. In addition, *MdSAUR60*, *MdSAUR62*, and *MdSAUR71* constituted three interleaved networks containing 20 mutual proteins and 144 individual proteins.

### 2.7. Protein–Protein Interaction Analysis of Candidate MdSAURs

The corresponding functional protein–protein interaction networks were reconstructed using the 25 candidate MdSAUR proteins to explore the functional protein–protein interactions and the gene regulatory relationships among these MdSAUR proteins (Figure 8). MdSAUR2, MdSAUR29, and MdSAUR71 proteins were regarded as crucial node proteins with big nodes and tight coordination for further research. In three specific proteins (or nodes), MdSAUR29 was the core hub-protein with big nodes and tight coordination in the mapped network. However, another 22 proteins had no interaction or connection to other proteins in the network.

### 2.8. Function Analysis of Primary Candidate MdSAURs

Through the network analysis and protein–protein interaction analysis, a total of seven SAURs were selected as primary candidate genes for regulating RGA formation for future study (Figure 6). *MdSAUR2*, *MdSAUR29*, *MdSAUR60*, and *MdSAUR62* had a higher expression level in stems of ‘BC’ than in those of ‘M9’ (Figure 3). *MdSAUR60*, *MdSAUR62*, *MdSAUR69* and *MdSAUR71* were mapped in QTL M10.1, and *MdSAUR84* was located in QTL B13.2 (Figure 4). The previous kompetitive allele-specific PCR (KASP) assay showed [25], the RGA of hybrids with the A:G genotype of marker Z3658 in QTL M10.1 were large, but had no significant difference from those with the A:A genotype (Appendix A). However, significantly larger RGA values were detected in hybrids with T:G than in hybrids with T:T genotype of the marker b13 in QTL B13.2 (Appendix A).

## 3. Discussion

### 3.1. MdSAUR Gene Family in Apple Genome GDDH13

In this present study, we successfully identified 96 *SAUR* genes in the new apple genome GDDH13, which has more SAUR genes and a more accurate prediction sequence than the old apple genome (80 members) [21,22]. Compared with the *SAUR* gene family in *Arabidopsis* (72), rice (58), and sorghum (71), many more members were identified in apple, suggesting that the *SAUR* family in apple experienced severe expansion during the long evolutionary history [26,27,28]. Some SAUR proteins were localized in the nucleus, cytoplasm, or plasma membrane [16,29,30]. Our results indicated that more than 97% of the SAUR proteins localized in the cytoplasm (Table 1). This feature has also been found in other species, such as cotton, watermelon, and maize [31,32,33]. SAUR clusters were reported in rice, tomato, and maize, in which most genes tended to be grouped together by phylogenetic analysis [27,31,34]. The SAUR proteins in the old apple genome were compared with those in *Arabidopsis* and rice, which constructed a phylogenetic tree containing six groups [21,22]. In our study, SAUR genes of the new apple genome GDDH13 were also grouped together with *Arabidopsis* and rice by phylogenetic analysis, which contained five groups and indicated that species specificity commonly existed in the SAUR family.

### 3.2. Differential Expression of MdSAUR Genes

The high expression divergence in different tissues reflected the complexity of gene family functions [35]. Given that the expression profiles of *SAUR* genes could provide important clues for gene function, the expression of *SAUR* genes in different tissues (leaf, stems, root, and flowers) were analyzed in maize, watermelon, and cotton [31,32,33]. Previous reports found that *SAUR* genes participating in the auxin signal pathway were upregulated or repressed to some extent following an auxin treatment [18,36]. In our study, expression profiles of *MdSAUR* genes were investigated in stems and roots after cutting by RNA-seq and qRT-PCR analysis. In fact, 61 *MdSAUR* genes were found to be expressed in the stems or roots, suggesting that the *MdSAUR* gene family might play a major role in the RGA development in apple. A total of 18 *MdSAUR* genes were found to have differential expression in stems, which acted as candidate genes for regulating RGA formation. Furthermore, the expression levels of *MdSAUR70* and *MdSAUR71* were high in stems, suggesting that they played key roles in RGA development. Similar to other primary auxin-responsive gene families (AUX/IAA, GH3), some members of SAUR gene family also exhibited tissue-specific expression patterns [34].

### 3.3. Allelic Variations and Gene Structures of MdSAUR Genes in QTLs

Auxin is essential for root growth and development, regulating stem cell specification and division, meristem size, cell elongation, and differentiation [37]. With *SAURs* being the most rapid auxin-responsive genes, most studies have focused on expression patterns and functional studies in shoots; several *SAUR* overexpression lines also exhibit root phenotypes [18]. Many QTLs of RGA have been reported in diverse species, such as sorghum (*Sorghum bicolor* L.), rice (*Oryza sativa* L.), and wheat (*Triticum aestivum* L.), which contributed to screening RGA by allelic variations [38,39,40]. In our study, 13 *MdSAURs* of QTL were screened by the allelic variations analysis based on the resequence database of ‘BC’ and ‘M9’, suggesting a likely genetic involvement of these *SAUR* genes in RGA formation.

The majority of *SAURs* lacked introns. For example, most of the *SAUR* genes in Solanaceae species possessed no introns and only 9 out of 99 *SAURs* in tomato and 3 out of 134 *SAURs* in potato had introns in their coding regions [34]. This phenomenon also exists in the *SAUR* genes of other species [28,33,41]. A similar phenomenon was found in the old apple genome, such that approximately 55% of the *SAUR* genes had no intron [22]. In our study, of 13 candidate *SAURs* in QTLs, only *MdSAUR69* had an intron in the new apple genome. As the occurrence of alternative splice in intronless genes is usually low, the function of certain *SAUR* family genes is likely stable.

### 3.4. Primary Candidate MdSAURs for RGA Formation

Network analysis of AppleMDO webtools could predict some proteins that positively or negatively regulate the target protein, which contributes to finding the primary target proteins with more regulatory proteins [25,42]. In our study, 25 *MdSAURs*, screened from expression and allelic variations, were further analyzed based on the network analysis of AppleMDO webtools. *MdSAUR2*, *MdSAUR60*, *MdSAUR62*, *MdSAUR69*, *MdSAUR71*, and *MdSAUR84* were screened as primary candidate genes because these *MdSAURs* regulated more proteins with PPC ≥ 9.0.

The STRING database aims to integrate all known and predicted associations between proteins, including both physical interactions and functional associations [43]. In the old apple genome, a total of 24 specific MdSAUR proteins were the core hub-proteins that were regarded as crucial nodes or core hub proteins with big nodes and tight coordination for further research [22]. In our study, these 25 key SAUR proteins were used to further identify the regulatory relationships with these putative functional protein–protein interactions using the STRING online database. Only three SAUR proteins (MdSAUR2, MdSAUR29 and MdSAUR71), which had big nodes and tight coordination in the mapped network, were selected as primary candidate genes.

Natural variation affects RGA formation, which has been verified in rice, corn, and wheat [43,44,45]. In our research, the allelic variations of MdSAURs (locatingin QTL) probably affected the gene function. The allelic variations of CDS in *MdSAUR62*, *MdSAUR69*, *MdSAUR71*, and *MdSAUR84* affected the amino acid sequences, and the allelic variants of MdSAUR62 upstream could result in the differential expression between ‘BC’ and ‘M9’ in stems. Furthermore, significantly larger RGA values were detected in hybrids with T:G than in hybrids with T:T genotype of the marker b13 in QTL B13.2.

## 4. Materials and Methods

### 4.1. Sequence Retrieval and Characterization Analysis

*MdSAUR* family numbers were achieved using the resequence database of ‘BC’ and ‘M9’ [25,46]. The locations and sequences of all *MdSAURs* were found via searching against the gene database of AppleMDO (http://bioinformatics.cau.edu.cn/AppleMDO/index.php (accessed on 22 October 2020) using the identity document (ID) of the gene [23]. Subcellular localization prediction of each of the *MdSAUR* family genes was carried out using the CELLO v2.5 server (http://cello.life.nctu.edu.tw/ (accessed on 22 October 2020) [47].

### 4.2. Phylogenetic Analysis in Arabidopsis, Rice, and Apple

Multiple sequence alignments for all available SAUR full-length protein sequences of *Arabidopsis* and apple were performed using the gene database of AppleMDO and TAIR (https://www.arabidopsis.org/index.jsp (accessed on 2 November 2020)). The rice SAUR genes were searched according to the previous report [27]. A neighbor-joining (NJ) phylogenetic tree of full-length sequences of SAURs in apple, rice, and *Arabidopsis* was constructed by the software MEGA 5.0 [48].

### 4.3. Gene Expression and qRT-PCR Analysis

In RNA-seq of hybrids from ‘BC’ and ‘M9’, the expression patterns of *MdSAURs* were analyzed at 7 d (stems) and 14 d (roots) after cutting [25]. To further characterize the expression of selected *MdSAUR* genes, tissue samples were also collected from ‘BC’ and ‘M9’ at 7 d (stems) and 14 d (roots) after cutting. Total RNA of stems and roots was extracted by using the CTAB method [49]. The first strand of cDNA was synthesized using the PrimerScript™ RT Reagent Kit with the gDNA Eraser (Perfect Real Time) kit from Takara (Tokyo, Japan). Specific primers were designed for the selected *MdSAUR* genes using Primer 5.0 software, and the actin gene of apple was used as a standardized internal reference (Appendix A). Expression levels of *MdSAURs* were determined by real-time quantitative PCR (RT-qPCR) [50]. Statistical analysis was performed using SPSS 17 software (Armonk, NY, USA).

### 4.4. Gene Structure and Allelic Variations Analysis

Mapchart 2.2 software was used to generate the chromosomal location image for these *MdSAUR* genes in QTL [51]. By comparing the resequencing database of ‘M9’ and ‘BC’, *MdSAUR* genes with SNPs or SVs, affecting cis-elements of the promoter and amino acids, were screened as candidate genes. The gene structures of these selected *MdSAURs* were found through the apple genome (GDDH13): JBrowse (https://www.rosaceae.org/tools/jbrowse (accessed on 1 December 2021) [23].

### 4.5. Network Analysis

Candidate *MdSAUR* genes, regulating RGA, were screened by the joint analysis of expression and allelic variation. To further screen the primary *MdSAUR* genes, network analysis of these candidate MdSAURs was analyzed using AppleMDO webtools [24,42]. Moreover, amino acid sequences of these MdSAURs were also used to analyze the protein–protein interactions using the STRING (https://cn.string-db.org/ (accessed on 12 May 2022)) [22].

### 4.6. Statistical Analysis

Expression differences of MdSAUR genes in stems and roots of ‘BC’ and ‘M9’ were analyzed by one-way analysis of variance (ANOVA) through Dunnett’s multiple comparison at a significance level of α = 0.05.

## 5. Conclusions

This study provides the phylogenetic analysis of the SAUR gene family in the new apple genome for the first time. In total, 25 MdSAURs regulating RGA were selected by the analysis of RNA-seq and BSA-seq. Then, combining the network analysis and protein–protein interaction analysis, seven SAURs (*MdSAUR2*, *MdSAUR29*, *MdSAUR60*, *MdSAUR62*, *MdSAUR69*, *MdSAUR71*, and *MdSAUR84*) were selected as primary genes for regulating RGA formation, to further study.

## Figures and Tables

**Figure 1 genes-13-02121-f001:**
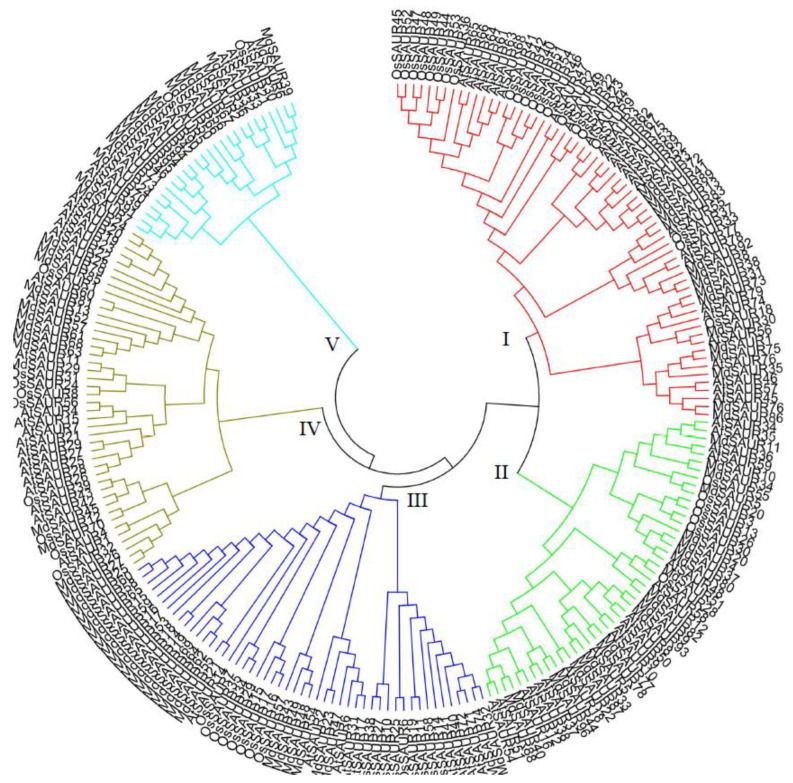
Phylogenetic tree of SAURs from *Arabidopsis*, rice and apple. To distinguish different subfamilies by the color of evolutionary branches.

**Figure 2 genes-13-02121-f002:**
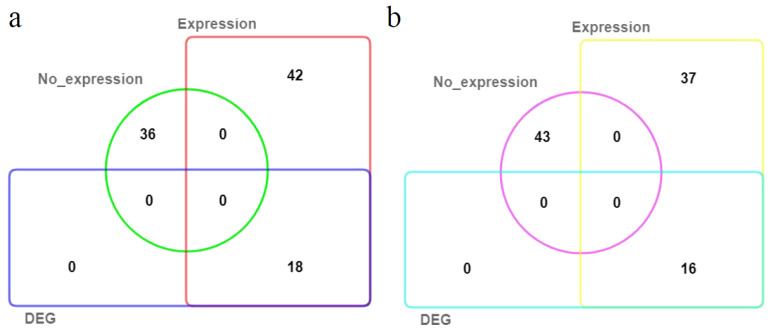
Venn diagrams showing overlap of *MdSAUR* expression in stems and roots. (**a**) Stems; (**b**) roots.

**Figure 3 genes-13-02121-f003:**
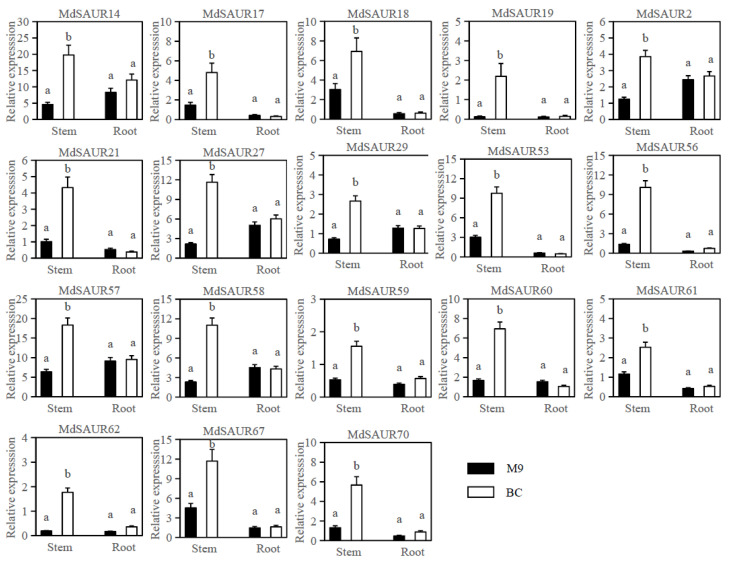
Differential expression profiles of 18 *MdSAURs* in stems and roots of ‘BC’ and ‘M9’. Expression analyses were performed by qRT-PCR. The black box represents the relative expression of *MdSAUR* genes in ‘M9’ and the white box in ‘BC’. Error bars represent the standard error of the average of three independent repeats. According to the LSD test, the different letters represent the significance between the paired means of *p* ≤ 0.05.

**Figure 4 genes-13-02121-f004:**
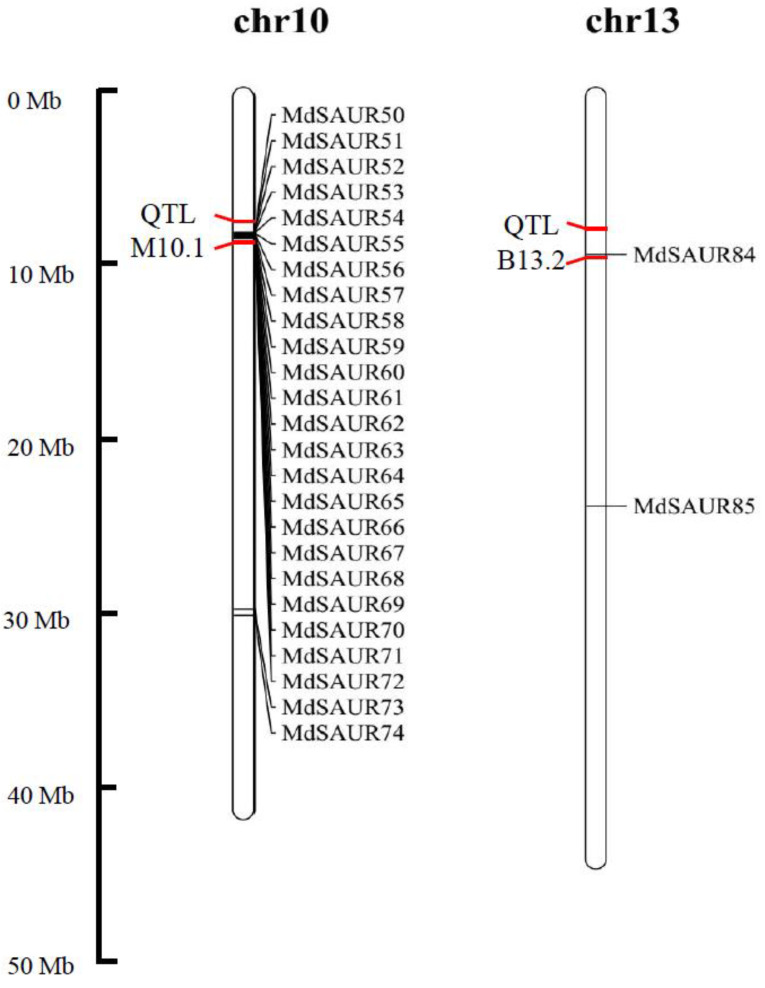
A co-localization analysis of SAUR genes with root growth angle (RGA) quantitative trait loci (QTL). Only genes co-localized with the RGA QTL are shown in the figure.

**Figure 5 genes-13-02121-f005:**
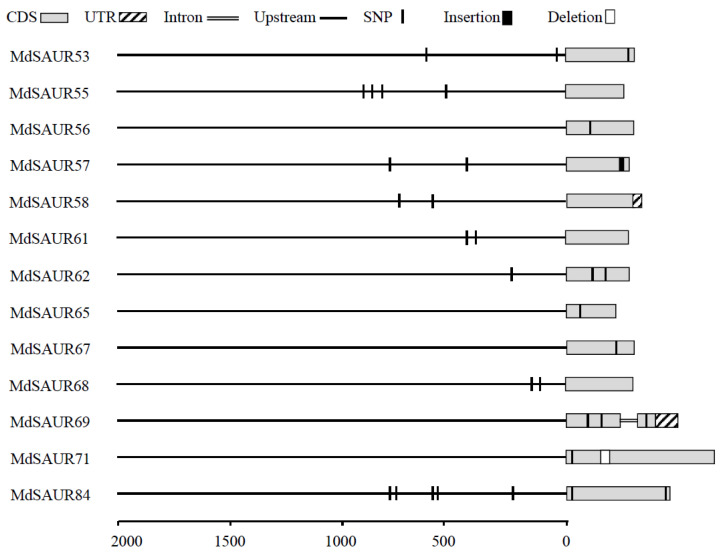
Gene structures and allelic variations prediction of candidate *MdSAUR* genes in QTL. Gene structures and allelic variations are represented by different shapes as indicated.

**Figure 6 genes-13-02121-f006:**
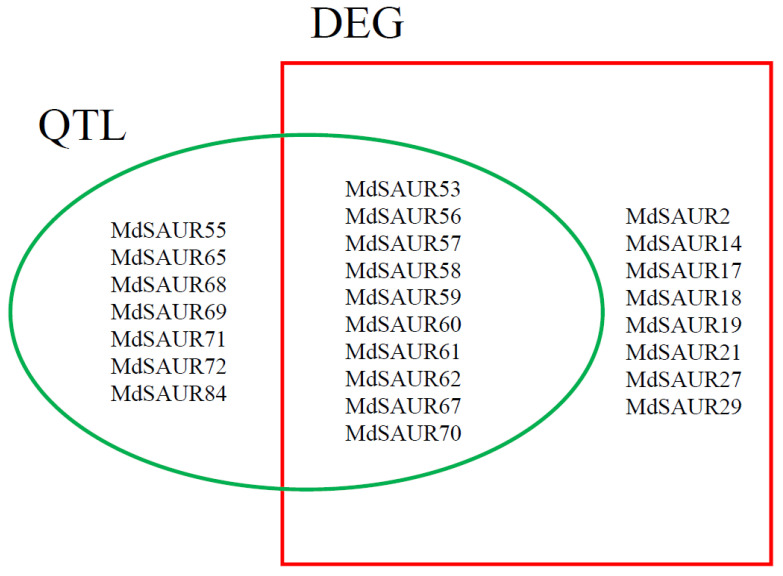
Candidate *MdSAUR* gene screening by expression and QTL. The blue font represents the primary candidate *MdSAURs*.

**Figure 7 genes-13-02121-f007:**
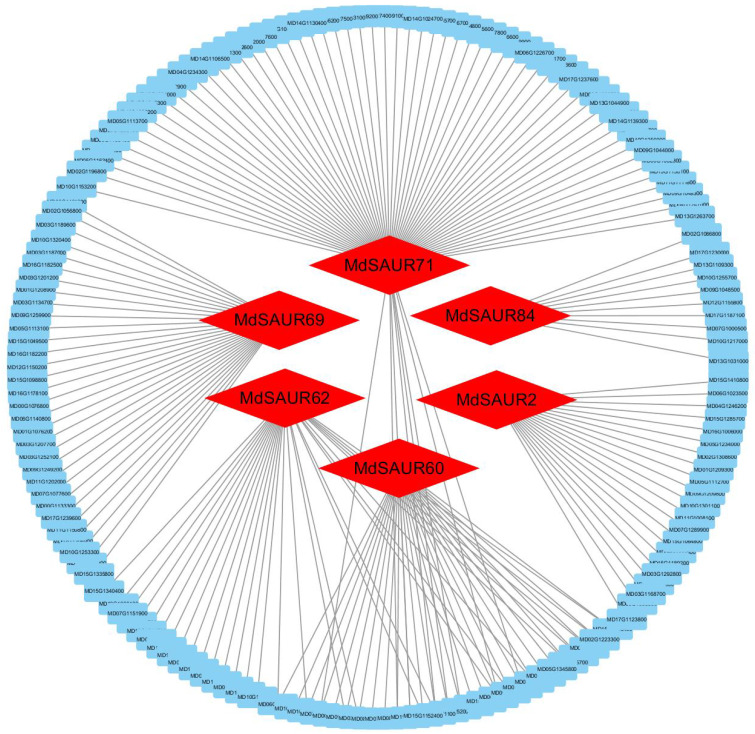
Cytoscape representation of co-expression between MdSAURs and regulatory proteins with Pearson correlation coefficient (PCC) ≥ 0.9 and the numbers of regulatory proteins ≥ 10. Red hexagons represent *MdSAURs*, blue rectangles represent regulatory proteins, and black lines represent potential regulatory relationships.

**Figure 8 genes-13-02121-f008:**
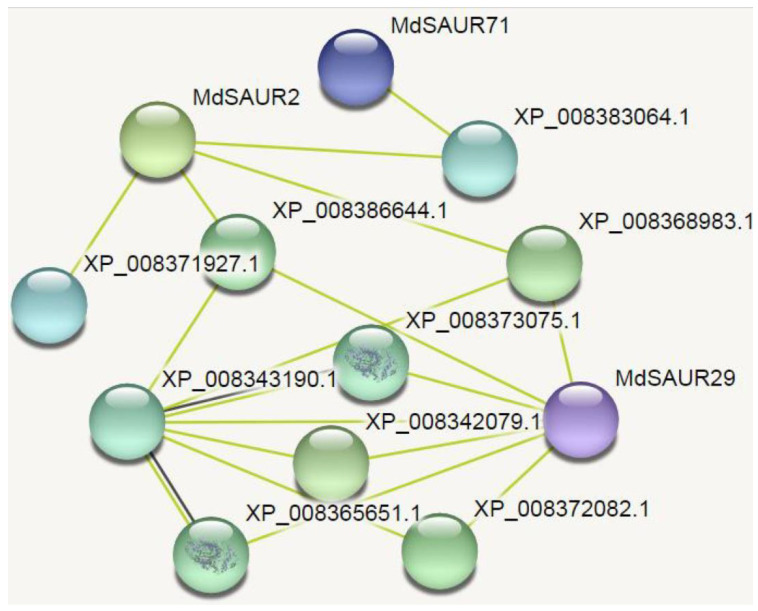
Mapped profile of protein–protein interaction network of candidate MdSAURs.

**Table 1 genes-13-02121-t001:** Small auxin-up RNA (SAUR) gene family in *Malus × domestica* genome GDDH13.

Name	Accession	Chromosome	Start	End	CDS (bp)	Protein Length (aa)	Location
MdSAUR1	MD00G1033700	Chr00	5801113	5801541	429	142	Cytoplasmic (1.899 *); Extracellular (1.209 *)
MdSAUR2	MD00G1036600	Chr00	6463961	6464332	372	123	Cytoplasmic (2.936 *)
MdSAUR3	MD00G1067200	Chr00	12921039	12921597	510	169	Cytoplasmic (2.078 *)
MdSAUR4	MD01G1106100	Chr01	21931954	21932349	396	131	Cytoplasmic (3.539 *)
MdSAUR5	MD01G1200100	Chr01	29699360	29699515	156	51	Cytoplasmic (2.608 *)
MdSAUR6	MD01G1227100	Chr01	31700591	31700992	402	133	Cytoplasmic (2.127 *); Periplasmic (1.436 *)
MdSAUR7	MD02G1100000	Chr02	7936729	7938938	531	176	Cytoplasmic (3.508 *)
MdSAUR8	MD02G1100200	Chr02	7980868	7981233	366	121	Cytoplasmic (3.490 *)
MdSAUR9	MD02G1133900	Chr02	10836759	10837079	321	106	Cytoplasmic (2.433 *)
MdSAUR10	MD02G1205700	Chr02	20662654	20663172	519	172	Cytoplasmic (1.888 *); Periplasmic (1.531 *)
MdSAUR11	MD04G1082600	Chr04	11472755	11473261	507	168	Cytoplasmic (2.682 *)
MdSAUR12	MD04G1093700	Chr04	17052592	17053047	456	151	Cytoplasmic (3.855 *)
MdSAUR13	MD04G1095800	Chr04	17659037	17659420	384	127	Cytoplasmic (3.689 *)
MdSAUR14	MD05G1051600	Chr05	8897921	8898226	306	101	Cytoplasmic (2.150 *)
MdSAUR15	MD05G1051700	Chr05	8916962	8917267	306	101	Cytoplasmic (2.102 *); Periplasmic (1.704 *)
MdSAUR16	MD05G1051800	Chr05	8919337	8919642	306	101	Cytoplasmic (1.714 *); Periplasmic (1.097 *)
MdSAUR17	MD05G1052000	Chr05	8924583	8924888	306	101	Cytoplasmic (1.678 *); Periplasmic (1.361 *)
MdSAUR18	MD05G1052100	Chr05	8941633	8941938	306	101	Cytoplasmic (2.424 *)
MdSAUR19	MD05G1052200	Chr05	8951907	8952203	297	98	Cytoplasmic (2.081 *)
MdSAUR20	MD05G1052300	Chr05	8953272	8953673	402	133	Cytoplasmic (1.127 *); OuterMembrane (1.262 *); Periplasmic (1.149 *)
MdSAUR21	MD05G1052400	Chr05	8999398	8999700	303	100	Cytoplasmic (1.615 *); OuterMembrane (1.235 *); Periplasmic (1.141 *)
MdSAUR22	MD05G1052500	Chr05	9013301	9013531	231	76	Cytoplasmic (2.429 *)
MdSAUR23	MD05G1052800	Chr05	9024586	9024882	297	98	Cytoplasmic (1.427 *); OuterMembrane (1.143 *); Periplasmic (1.023 *)
MdSAUR24	MD05G1053200	Chr05	9075063	9075681	438	145	Cytoplasmic (1.306 *); Periplasmic (1.843 *)
MdSAUR25	MD05G1113200	Chr05	23165138	23165296	159	52	Cytoplasmic (2.493 *)
MdSAUR26	MD05G1113300	Chr05	23178916	23179197	282	93	Cytoplasmic (2.159 *)
MdSAUR27	MD05G1113400	Chr05	23191697	23191981	285	94	Cytoplasmic (1.729 *); Periplasmic (1.323 *)
MdSAUR28	MD05G1113500	Chr05	23236666	23236950	285	94	Cytoplasmic (2.288 *)
MdSAUR29	MD05G1113600	Chr05	23245653	23245961	309	102	Cytoplasmic (2.931 *)
MdSAUR30	MD05G1113700	Chr05	23253593	23253868	276	91	Cytoplasmic (3.139 *)
MdSAUR31	MD05G1219800	Chr05	35019487	35019882	396	131	Cytoplasmic (2.886 *)
MdSAUR32	MD05G1223400	Chr05	35609400	35609819	420	139	Cytoplasmic (3.689 *)
MdSAUR33	MD06G1137300	Chr06	28237372	28237890	519	172	Periplasmic (2.074 *)
MdSAUR34	MD07G1117400	Chr07	14414803	14415318	516	171	Cytoplasmic (1.078 *); Extracellular (1.091 *); OuterMembrane (1.143 *); Periplasmic (1.413 *)
MdSAUR35	MD07G1151400	Chr07	22071122	22071418	297	98	Cytoplasmic (1.496 *); Periplasmic (1.048 *)
MdSAUR36	MD07G1156400	Chr07	22734374	22734730	357	118	Cytoplasmic (3.157 *)
MdSAUR37	MD07G1172500	Chr07	25001027	25001422	396	131	Cytoplasmic (3.712 *)
MdSAUR38	MD07G1220700	Chr07	29829638	29830003	366	121	Cytoplasmic (2.399 *)
MdSAUR39	MD07G1297400	Chr07	35727069	35727473	405	134	Cytoplasmic (2.401 *)
MdSAUR40	MD08G1014900	Chr08	1133020	1133337	318	105	Cytoplasmic (3.380 *)
MdSAUR41	MD08G1015100	Chr08	1172890	1173342	453	150	Cytoplasmic (2.966 *)
MdSAUR42	MD08G1124000	Chr08	11654816	11655337	522	173	Cytoplasmic (2.014 *)
MdSAUR43	MD09G1089100	Chr09	6500577	6500903	327	108	Cytoplasmic (3.287 *)
MdSAUR44	MD09G1166200	Chr09	13620973	13621542	570	189	Cytoplasmic (2.487 *)
MdSAUR45	MD09G1166300	Chr09	13625736	13626182	447	148	Cytoplasmic (2.873 *)
MdSAUR46	MD09G1166400	Chr09	13644321	13644788	468	155	Cytoplasmic (1.905 *); Periplasmic (1.485 *)
MdSAUR47	MD09G1170600	Chr09	14396726	14397252	399	132	Cytoplasmic (2.009 *)
MdSAUR48	MD09G1176000	Chr09	15024607	15024960	354	117	Cytoplasmic (2.704 *)
MdSAUR49	MD09G1176100	Chr09	15030105	15030461	357	118	Cytoplasmic (2.406 *); Periplasmic (1.898 *)
MdSAUR50	MD10G1058700	Chr10	7939190	7939543	354	117	Cytoplasmic (4.012 *)
MdSAUR51	MD10G1059000	Chr10	7963586	7964688	192	63	Cytoplasmic (2.493 *)
MdSAUR52	MD10G1059100	Chr10	7965729	7966221	219	72	Cytoplasmic (1.404 *); Periplasmic (1.922 *)
MdSAUR53	MD10G1059200	Chr10	7969469	7969774	306	101	Cytoplasmic (1.674 *); Periplasmic (1.907 *)
MdSAUR54	MD10G1059300	Chr10	7980413	7980586	174	57	Cytoplasmic (2.033 *)
MdSAUR55	MD10G1059400	Chr10	7982312	7982581	270	89	Cytoplasmic (1.921 *); OuterMembrane (1.373 *)
MdSAUR56	MD10G1059500	Chr10	7993321	7993620	300	99	OuterMembrane (1.274 *); Periplasmic (1.751 *)
MdSAUR57	MD10G1059600	Chr10	8008946	8009221	276	91	Cytoplasmic (1.296 *); Extracellular (1.015 *); Periplasmic (1.233 *)
MdSAUR58	MD10G1059700	Chr10	8021965	8022290	306	101	Cytoplasmic (2.413 *)
MdSAUR59	MD10G1059800	Chr10	8024227	8026563	432	143	Cytoplasmic (1.674 *); InnerMembrane (1.774 *)
MdSAUR60	MD10G1060100	Chr10	8086085	8086390	306	101	Cytoplasmic (2.139 *)
MdSAUR61	MD10G1060200	Chr10	8087982	8088257	276	91	Cytoplasmic (1.558 *); OuterMembrane (1.341 *)
MdSAUR62	MD10G1060300	Chr10	8120383	8120679	297	98	Cytoplasmic (2.887 *)
MdSAUR63	MD10G1060400	Chr10	8121964	8122149	186	61	Cytoplasmic (1.475 *); Periplasmic (1.727 *)
MdSAUR64	MD10G1060500	Chr10	8123002	8126545	489	162	Cytoplasmic (1.570 *); OuterMembrane (1.282 *);
MdSAUR65	MD10G1060600	Chr10	8150440	8150652	213	70	Cytoplasmic (1.829 *); Periplasmic (1.462 *)
MdSAUR66	MD10G1060700	Chr10	8164264	8164668	405	134	Cytoplasmic (1.768 *); Periplasmic (1.637 *)
MdSAUR67	MD10G1060800	Chr10	8165745	8166044	300	99	Cytoplasmic (1.393 *); OuterMembrane (1.392 *); Periplasmic (1.321 *)
MdSAUR68	MD10G1060900	Chr10	8171485	8171760	276	91	Cytoplasmic (2.003 *)
MdSAUR69	MD10G1061000	Chr10	8179736	8180225	324	107	Cytoplasmic (2.279 *)
MdSAUR70	MD10G1061100	Chr10	8207917	8208257	318	105	Cytoplasmic (2.723 *)
MdSAUR71	MD10G1061300	Chr10	8240210	8240869	660	219	Periplasmic (2.269 *)
MdSAUR72	MD10G1061400	Chr10	8286787	8287227	441	146	Cytoplasmic (3.600 *)
MdSAUR73	MD10G1202100	Chr10	29998840	29999244	405	134	Cytoplasmic (3.237 *)
MdSAUR74	MD10G1204800	Chr10	30376680	30377099	420	139	Cytoplasmic (3.504 *)
MdSAUR75	MD11G1145300	Chr11	13514349	13514926	474	157	Cytoplasmic (1.870 *); Extracellular (1.259 *); OuterMembrane (1.527 *)
MdSAUR76	MD12G1027600	Chr12	3084745	3085128	384	127	Cytoplasmic (2.781 *)
MdSAUR77	MD12G1088000	Chr12	10835118	10835579	462	153	Cytoplasmic (1.851 *)
MdSAUR78	MD12G1088100	Chr12	10872715	10873179	465	154	Cytoplasmic (1.361 *); OuterMembrane (1.856 *)
MdSAUR79	MD12G1088200	Chr12	10875190	10875582	393	130	Cytoplasmic (1.296 *); Extracellular (1.450 *); OuterMembrane (1.699 *)
MdSAUR80	MD12G1097400	Chr12	15303943	15304317	375	124	Cytoplasmic (2.793 *)
MdSAUR81	MD12G1098500	Chr12	15404672	15404980	309	102	Cytoplasmic (2.557 *)
MdSAUR82	MD12G1113400	Chr12	18052351	18052800	450	149	Cytoplasmic (3.389 *)
MdSAUR83	MD12G1115800	Chr12	18457034	18457417	384	127	Cytoplasmic (3.747 *)
MdSAUR84	MD13G1123500	Chr13	9172512	9172973	462	153	Cytoplasmic (2.300 *); Periplasmic (2.108 *)
MdSAUR85	MD13G1235900	Chr13	23824202	23824513	312	103	Cytoplasmic (2.307 *)
MdSAUR86	MD14G1028000	Chr14	2503144	2503530	387	128	Cytoplasmic (2.861 *)
MdSAUR87	MD14G1152100	Chr14	24578067	24578540	474	157	Cytoplasmic (1.569 *); Periplasmic (1.974 *)
MdSAUR88	MD15G1014200	Chr15	805247	805606	318	105	Cytoplasmic (3.406 *)
MdSAUR89	MD15G1014300	Chr15	821989	822441	453	150	Cytoplasmic (3.047 *)
MdSAUR90	MD15G1102700	Chr15	7267646	7268164	519	172	Cytoplasmic (2.378 *)
MdSAUR91	MD15G1222900	Chr15	18103292	18105613	555	184	Cytoplasmic (3.251 *)
MdSAUR92	MD15G1223100	Chr15	18137898	18138260	363	120	Cytoplasmic (2.878 *)
MdSAUR93	MD15G1246800	Chr15	20468388	20468708	321	106	Cytoplasmic (3.366 *)
MdSAUR94	MD16G1124300	Chr16	8987819	8988280	462	153	Cytoplasmic (2.104 *); Periplasmic (2.320 *)
MdSAUR95	MD16G1240600	Chr16	26000543	26001037	495	164	Cytoplasmic (2.502 *)
MdSAUR96	MD17G1161400	Chr17	15886464	15887039	576	191	Cytoplasmic (2.238 *)

## Data Availability

All RNA-seq reads and BSA-seq data have been deposited in the NCBI SRA under the accession number PRJNA655783. The DNA resequencing reads of ‘Baleng Crab’ and ‘M9’ are freely available in the NCBI SRA under the accession numbers SRR12234087 and SRR12233711, respectively. The data presented in this study are available in the graphs and tables provided in the manuscript.

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
