# Peer review of "Analysis of the Small Auxin-Up RNA (SAUR) Genes Regulating Root Growth Angle (RGA) in Apple"

_genes, 2022, doi:10.3390/genes13112121_

Round 1
Reviewer 1 Report
The MS was reviewed thoroughly, found according to journal format.
all parts are presented very well. the science knowledge is sound and full the MS title.
Results interpreted in a good way. discussion section matching the results and self explanatory. but still there is need of making few improvements.
the English language is not upto the mark for readers. so the MS should be revised from English language expert.
conclusion section may improved by adding more novelity found in the MS.
References should be cross checked.
it is strongly recommended to add recent citation
Author Response
Reviewer 1
The MS was reviewed thoroughly, found according to journal format. all parts are presented very well. the science knowledge is sound and full the MS title. Results interpreted in a good way. discussion section matching the results and self explanatory. but still there is need of making few improvements. the English language is not upto the mark for readers. so the MS should be revised from English language expert. conclusion section may improved by adding more novelity found in the MS. References should be cross checked. it is strongly recommended to add recent citation.
Comment 1: Discussion section matching the results and self explanatory. but still there is need of making few improvements.
Authors’ response: We appreciate this comment and few improvements revision has been made accordingly.
Comment 2: English language is not upto the mark for readers. so the MS should be revised from English language expert.
Authors’ response: We accepted the reviewer’s suggestion and the manuscript was re-edited by a English language editing service.
Comment 3: Conclusion section may improved by adding more novelity found in the MS.
Authors’ response: We agree with the reviewer and the conclusion section has been re-organized.
Comment 4: References should be cross checked. it is strongly recommended to add recent citation.
Authors’ response: Thanks for the comment. References have been cross checked, and we also added some recent citations.

Reviewer 2 Report
I have gone through the manuscript entitled "Analysis of the small auxin-up RNA (SAUR) genes regulating root growth (RGA) angle in apple". In this manuscript the authors have identified 96 Malus domestics Small auxin up-regulated RNAs through out the genome. Of these identified Small auxin up-regulated RNAs 25 were tested for their role as genetic variant and QTL for root growth angle. Finally some important Small auxin up-regulated RNAs viz., MdSAUR2, MdSAUR29, MdSAUR60, MdSAUR62 were functionally validated for their role in root growth angle. The experiment is well designed and objectives were conducted well however I suggest the authors to show the figure of root growth angle(high or low) with the impact of these identified and validated candidate genes. Additionally the conclusion section needs more attention.
Author Response
Reviewer 2
I have gone through the manuscript entitled "Analysis of the small auxin-up RNA (SAUR) genes regulating root growth (RGA) angle in apple". In this manuscript the authors have identified 96 Malus domestics Small auxin up-regulated RNAs through out the genome. Of these identified Small auxin up-regulated RNAs 25 were tested for their role as genetic variant and QTL for root growth angle. Finally some important Small auxin up-regulated RNAs viz., MdSAUR2, MdSAUR29, MdSAUR60, MdSAUR62 were functionally validated for their role in root growth angle. The experiment is well designed and objectives were conducted well however I suggest the authors to show the figure of root growth angle(high or low) with the impact of these identified and validated candidate genes. Additionally the conclusion section needs more attention.
Comment 1: I suggest the authors to show the figure of root growth angle(high or low) with the impact of these identified and validated candidate genes.
Authors’ response: We accepted the reviewer’s suggestion and the revised accordingly in result section.
Comment 2: The conclusion section needs more attention.
Authors’ response: We agree with the reviewer and the conclusion section has been re-organized.

Reviewer 3 Report
The purpose of this study is to determine whether small auxin up-regulated RNAs (SAURs) play a specific role in controlling the root growth angle (RGA) of apple rootstock. The study's findings are quite intriguing, with the key candidate genes for controlling RGA being identified as MdSAUR2, MdSAUR29, MdSAUR60, MdSAUR62, MdSAUR69, MdSAUR71, and MdSAUR84. Overall, the study question is current, and the introduction portion of this publication gives the aims enough context. The experimental design and research approach are appropriate. The results are presented in the manuscript in a very clear and succinct manner, and the writing is excellent. The discussion portion provided a complete and excellent response to the query posed in the introductory section, along with a strong defense of the findings. Overall, I must praise the authors on their superb effort.
Author Response
Reviewer 3
The purpose of this study is to determine whether small auxin up-regulated RNAs (SAURs) play a specific role in controlling the root growth angle (RGA) of apple rootstock. The study's findings are quite intriguing, with the key candidate genes for controlling RGA being identified as MdSAUR2, MdSAUR29, MdSAUR60, MdSAUR62, MdSAUR69, MdSAUR71, and MdSAUR84. Overall, the study question is current, and the introduction portion of this publication gives the aims enough context. The experimental design and research approach are appropriate. The results are presented in the manuscript in a very clear and succinct manner, and the writing is excellent. The discussion portion provided a complete and excellent response to the query posed in the introductory section, along with a strong defense of the findings. Overall, I must praise the authors on their superb effort.
Authors’ response: We appreciate the positive comments from the reviewer.
